# Quality of Life Factors in Adults with Eosinophilic Oesophagitis in New Zealand

**DOI:** 10.3390/nu16203437

**Published:** 2024-10-10

**Authors:** Vicki McGarrigle, Akhilesh Swaminathan, Stephen James Inns

**Affiliations:** 1Department of Gastroenterology, Hutt Valley Hospital, Wellington 5010, New Zealand; 2Department of Gastroenterology, Middle Hospital, Auckland 6021, New Zealand; 3Department of Medicine, University of Otago, Christchurch 8011, New Zealand; akhilesh.swaminathan@otago.ac.nz; 4Department of Medicine, University of Otago, Wellington 6021, New Zealand

**Keywords:** eosinophilic oesophagitis, quality of life

## Abstract

Background: Eosinophilic oesophagitis (EoE) is an immune-mediated oesophageal disorder causing dysphagia. Patients with EoE experience reduced QoL due to symptoms; however, this has not been assessed in the New Zealand population. The aim of this study was to assess QoL in patients with EoE in New Zealand. Methods: This observational study recruited participants from two New Zealand hospitals. Records were reviewed to confirm diagnoses, and consenting participants completed an electronic survey, consisting of the Dysphagia Symptom Questionnaire (DSQ) score and the QoL-specific EoE (EoE-QoL-A) questionnaire score. Correlation analysis examined the relationship between the DSQ and EoE-QoL-A scores. Differences in baseline variables were assessed. Univariate logistic regression assessed the association of variables with disease activity and QoL. Results: Fifty-four participants responded, and four were excluded due to incomplete surveys. The majority (76%) were male, and the median age was 47 years (IQR 42–58). The median DSQ was 49 (IQR 0–60), and the median EoE-QoL-A score was 68 (IQR 48–80). A reduced EoE-QoL-A score was associated with active disease (OR = 0.96,95% CI 0.926–0.995). Significant associations were found between disease activity and overall EoE-QoL-A score (r = −0.37, *p* < 0.01) as well as the sub-categories eating and diet (r = −0.54, *p* < 0.001), social (r = 0.30, *p* < 0.05), and emotional impact (r = −0.44, *p* < 0.01). The EOE-QoL-A score was higher in those on PPI (75 vs. 60, *p* = 0.02). Conclusion: This study identified a decreased quality of life (QoL) in individuals with EoE in New Zealand, aligning with international literature. The increased DSQ scores suggest a possible gap in current management approaches. The correlation between the DSQ and QoL highlights the need for improved care models of care for EoE.

## 1. Introduction

Eosinophilic oesophagitis (EoE) is an immune-mediated chronic inflammatory disease of the oesophagus [1,2], first characterised in the early 1990s. The prevalence of EoE has significantly increased, with some countries reporting p to 131-fold increase in the past 14 years [3]. The prevalence of EoE varies internationally, with rates between 22.7 and 58.9 per 100,000 persons [3,4,5]. A study on EOE in New Zealand showed an annual incidence of 6.95 per 100,000 persons/year [6], notably higher than rates reported in Europe and America at that time [4,7]. A 2023 global incidence systematic review and meta-analysis showed an incidence rate of 5.31 cases per 100,000 persons-years, again lower than the known incidence rate in New Zealand [8].

Oesophageal dysfunction in EoE occurs because of remodelling due to chronic eosinophilic inflammation, which contributes to symptoms of dysphagia and food impaction [9,10,11]. The severity of the symptoms of EoE correlates with oesophageal eosinophilia, endoscopic appearance [12,13,14], and quality of life (QoL) [2,15,16]. A systematic review in 2014 by Mukkada et al. including 22 studies showed that active EoE significantly reduced QoL, and in turn, the treatment of the disease significantly improved QoL. The impact of EoE on QoL is largely related to the degree of symptoms [15]; however, this has not been assessed in the New Zealand population. 

Health-related QoL can be measured using generic QoL assessment tools; however, we know that disease-specific QoL assessments are more sensitive to changes in disease status than generic tools are [17,18], as seen in the inflammatory bowel disease population [19]. The Adult Eosinophilic Oesophagitis QoL (EoO-QOL-A) questionnaire is a validated EoE-specific QoL measure that has been used in European and American populations [20,21,22,23,24,25]. Studies have demonstrated a correlation between EoE QoL and symptom severity, endoscopic findings, and histological disease activity [15,21,22,26]. 

Importantly, disease activity is not the only factor attributed to reduced quality of life in patients with EoE. Multiple studies have demonstrated that food bolus obstruction significantly impacts QoL [21,27]. In a more recent multi-centre study, food bolus obstruction, higher level of education, dietary restriction, and disease duration were found to be the strongest predictors of reduced quality of life in EoE, as assessed by the EoE-QoL-A, with food impaction affecting all domains [21]. The literature regarding the influence of gender on QoL is inconsistent, with some studies indicating that females with EoE experience a lower QoL [21,26,28] and others reporting no significant difference [29]. Interestingly, although specific dietary restrictions can improve EoE disease activity [30,31], they are also associated with reduced QoL in EoE [16,17,21,32].

This study aimed to assess the QoL and the determinants of QoL in individuals with EoE in New Zealand, focusing on the active and inactive disease states of EoE. We aimed to determine whether the impact of EoE on QoL is similar to that described in international populations. 

### The Study Hypothesised That

Individuals who report active symptoms of EoE, as defined by a DSQ score ≥ 14, have a lower QoL compared with those with inactive disease (DSQ < 14)Factors such as dietary restrictions, food bolus obstruction, and female gender are associated with lower QoL, as defined by an EoE-QoL-A score less than the median score.

## 2. Methods

### 2.1. Study Participants

This observational cohort study recruited adult participants (aged ≥ 18 years) with an established diagnosis of EoE [2,33] from two major healthcare facilities in New Zealand: Middlemore Hospital, Auckland, and Hutt Valley Hospital, Wellington. Eligible participants had a confirmed diagnosis of EoE based on endoscopic and histological findings and symptoms of dysphagia.

The diagnosis of EoE was verified by the presence of a previous endoscopic examination containing the eosinophilic oesophagitis endoscopic reference score (EREFS), which is a standard form of EoE assessment that reports the presence of oesophageal oedema, rings, exudates, furrows, and strictures [34]. Each feature is scored according to its severity, thus providing an overall measurement of disease severity. Additionally, EoE diagnosis was confirmed by the presence of >15 eosinophils per high power field (hpf) on an oesophageal biopsy. The indication for each endoscopy was reviewed to ensure the presence of relevant symptoms of dysphagia or food bolus impaction, as oesophageal eosinophilia can occur in up to 5% of asymptomatic individuals, with >15 eosinophils per hpf identified in 1.1% of the general population [35]. 

Suitable participants were identified using two distinct methods. In the Hutt Valley cohort, individuals were identified using a pre-existing database of patients diagnosed with EoE between January 2011 and December 2021. For the Auckland cohort, participants were identified through a database search of endoscopic examinations performed at Middlemore Hospital between January 2011 and December 2021 (Provation, Minneapolis, MN, USA) using the search term eosinophilic oesophagitis. Records were thoroughly reviewed to confirm the presence of ≥15 eosinophils per hpf on oesophageal biopsy and ensure that the diagnosis met the diagnostic criteria [33]. Individuals identified to have a diagnosis of EoE were subsequently approached for participation in this study. Study participants’ electronic medical records were extensively examined for details on patient demographics, comorbidities (including atopy), medications, EoE therapies, hospital admissions for food bolus obstructions, and previous endoscopic examinations. Participants completed an electronic survey questionnaire that consisted of the Dysphagia Symptom Questionnaire (DSQ) and the EoE Disease-Specific QoL (EoE-QoL-A) questionnaire using REDCap web application (Research Electronic Data Capture, REDCap, Vanderbilt, Nashville, TN, USA).

### 2.2. Study Related Questionnaires

EoE symptom activity was recorded using the DSQ. The DSQ is a validated and commonly used patient-reported outcome measure (PROM) that uses a 4-point scale to assess the frequency of dysphagia and degree of dysphagia severity associated with EoE [36]. Participants recalled and recorded the frequency and severity of symptoms over the prior 14 days. Higher DSQ scores are associated with worse symptoms. Participants with a total DSQ score of ≥14 were classified as having active disease, while those with a score of <14 were categorized as having inactive disease. The maximum possible daily DSQ score is 6, and the overall DSQ score is calculated as the average daily score over 14 consecutive days (maximum score 84). A daily DSQ score of ≥1 reflects the presence of symptoms; thus, a total score of <14 was used to distinguish between active and inactive disease. Although the DSQ does not have established reference ranges to define active disease, a reduction of 13.5 points in the DSQ score represents an improvement in symptoms [37]. Therefore, dichotomization using our threshold of 14 was considered appropriate. 

EoE QoL was assessed using the EoE-QOL-A, a 37-point questionnaire that assesses QoL separated into multiple domains: eating/diet impact, social impact, emotional impact, disease anxiety, and choking anxiety [38]. Higher EoE-QOL-A total and sub-scale scores are associated with better QoL (Table 1). EoE-QOL-A scores were determined using the methodology proposed by Taft et al. [38,39]; however, there are no established thresholds on the EoE-QOL-A that signify an impaired QoL. The median EoE-QoL-A score in this cohort was used to then dichotomise the scores into impaired (<median score) and not-impaired (≥median score).

All de-identified study data were stored in a secure database (Research Electronic Data Capture, REDCap, Vanderbilt, Nashville, TN, USA).

### 2.3. Ethical Approval

This study was granted full approval by the Human Ethics Committee of the University of Otago Medical School under the reference number H22/057. All procedures performed in the study were in accordance with the ethical standards of the institutional research committee and with the 1964 Helsinki Declaration and its later amendments or comparable ethical standards. Informed consent was obtained from all individual participants included in the study.

### 2.4. Statistical Analyses

Baseline study variables were reported using median (IQR) and proportions (%). Spearman’s rank correlations were used to examine the associations between continuous variables, including the DSQ and EoE-QOL-A and each of its subscales. The Mann–Whitney *U* test was used to assess differences in continuous variables between groups. Fisher’s exact test was used to assess differences in categorical variables between groups. These non-parametric tests were used due to the non-normal distribution of the EoE-QoL-A scores found using the Shapiro–Wilk test. Median dichotomisation was performed on the EoE-QOL-A to identify those with an impaired QoL (total EoE-QOL-A score below median) and unimpaired QoL (total EoE-QOL-A score ≥ median). Univariable logistic regression was used to determine baseline variables associated with active EoE symptoms (DSQ ≥ 14) and impaired QoL (EoE-QOL-A < median). 

Statistical analyses were performed using SPSS 27 (IBM Corp., Armonk, NY, USA), and graphs of the study data were produced using GraphPad Prism 9 (GraphPad Software Inc., San Diego, CA, USA). This study was performed following approval by the University of Otago Human Ethics Committee (Health) and in accordance with the World Medical Assembly Declaration of Helsinki.

## 3. Results

A total of 116 individuals were identified who met the study inclusion criteria. Of these, 93 were successfully contacted, and 50 completed all study surveys (Figure 1). 

The 50 participants included in the study analyses had a median age of 47 years (IQR 42–58), and 11 (22%) were female. Active EoE symptoms using the DSQ score were reported in 31 (62%). Two-thirds of participants had a previous hospital admission for food bolus obstruction (66%), and 60% were using proton pump inhibitor (PPI) therapy for their EoE (Table 2).

### Associations between Study Variables and EoE Disease Activity

In this cohort, the presence of active EoE was not associated with age (median age: active EoE = 50 years (IQR 43–60) vs. inactive EoE = 43 years (IQR 40–57); *p* = 0.28), gender (female: active EoE = 9/30 vs. inactive EoE = 2/17, *p* = 0.28), history of atopy (active EoE = 4/31 vs. inactive EoE = 4/19, *p* = 0.46), or history of food bolus obstruction (active EoE = 21/31 vs. inactive EoE = 12/19, *p* = 0.77). However, PPI use was significantly lower in those with active EoE than in those with inactive EoE (15/31 vs. 15/19, *p* = 0.04) (Table 3).

A significant association was noted between active EoE (DSQ ≥ 14) and impaired QoL (EoE-QoL-A < 68) (OR = 0.96, 95% CI 0.926–0.995). The total EoE-QOL-A score and the subscale scores for eating/diet and emotional impact were significantly lower in those with active EoE compared with those with inactive EoE (*p* < 0.05) (Figure 2). 

The median EoE-QOL-A score in this cohort was 68 (IQR 48–80) (Table 2). The overall EoE-QOL-A scores were negatively correlated with the DSQ score (r = −0.37, *p* < 0.01). The subscales of the EoE-QoL-A, including eating/diet (r = −0.54, *p* < 0.001), social impact (r = −0.30, *p* < 0.05), and emotional impact (r = −0.44, *p* < 0.01), were also significantly correlated with DSQ scores in all participants (Table 4). 

EoE QoL scores were significantly lower in those with active EoE (DSQ ≥ 14) (active EoE, median EoE-QOL-A = 59, IQR 46–76; inactive EoE, median EoE-QOL-A = 75, IQR 62–89; *p* = 0.02) (Table 5). EoE QoL scores were significantly higher in individuals on PPI therapy (on PPI, median EoE-QOL-A = 75, IQR 53–83; not on PPI, median EoE-QOL-A = 60, IQR 45–74; *p* = 0.02). 

Univariable logistic regression analyses (Table 6) showed that active EoE (DSQ ≥ 14) was the only factor significantly associated with impaired EoE QoL, as defined by an EoE-QoL-A score less than 68 (OR = 1.02, 95% CI 1.002–1.04). A two-tailed correlation t-test showed a significantly lower EoE-QoL-A score in those with active disease between our study and Taft’s original study (mean difference −20.9 *p* < 0.001, 95% CI −29.8 to −12.1). 

## 4. Discussion

This is the first study to investigate the QoL of patients with EoE in New Zealand. We demonstrated a significant reduction in QoL in patients with EoE, which relates to EoE disease activity. Patients on PPI treatment had lower symptom scores and better quality of life compared with those not on PPI treatment.

Health-related QoL is an important outcome measure, and the use of disease-specific QoL tools is important in providing information on specific disease-related factors contributing to QoL that are not assessed by generic measures. The EoE-QoL-A is a measure specific to EoE created in 2011 by Taft et al. Taft’s original validation study involving 201 patients with EoE showed a reduced QoL in those with active disease [38]. Our cohort reported lower QoL scores compared with Taft’s original validation study, suggesting a greater QoL impact in patients with EoE in New Zealand. This may reflect the limitations of healthcare access, such as long endoscopy wait times, which can lead to further complications [40].

Interestingly, food bolus obstruction was noted in 66% of our cohort, higher than the 16–20% [41,42] reported in the literature, highlighting the use of resources due to EoE complications. Food bolus in those with EoE is related to chronic untreated inflammation leading to fibrosis [43], and a long duration of untreated eosinophilia is a risk factor for stricturing disease [9]. Recurrent food bolus has been strongly correlated with reduced QoL in EoE [21]; however, we did not find any association in this cohort. The observed high rates of active EoE and the frequent occurrence of food bolus episodes highlight a pressing need to re-evaluate the current standards of care and management for patients with EoE in New Zealand.

While elimination diets are a proven treatment in EoE [44], only 6% of our cohort were following an elimination diet, which may have been due to patient choice, physician factors, or a lack of dietic resources. Elimination diets are not suitable for everyone due to, e.g., busy lifestyles, other required elimination diets, and restrictive eating habits, but they should be discussed with appropriate patients, as they may mitigate the need for long-term medications [12,16,35,45].

This study, while pioneering in its examination of the quality of life (QoL) in patients with EoE in New Zealand, has several limitations that warrant attention:Sample Size: Our sample size of 50 participants, although sufficient for preliminary insights, is relatively small and may restrict the generalisability of our findings.Absence of Set Normative Values for QoL and DSQ Scores: A significant challenge in our analysis was the lack of established normative values for the QoL and DSQ scores used. Their absence in our study may hinder the accurate interpretation and contextual understanding of our findings.Dichotomisation of DSQ and EoE-QoL-A Scores: The dichotomisation of the DSQ score in our study introduces a risk of misclassification, potentially skewing the correlation with QoL outcomes. This methodological approach may overlook subtle variations in disease severity and its impact on QoL. Similarly, the dichotomisation of EoE-QoL-A scores could simplify complex data, potentially overlooking critical aspects of patient experiences. Future research could explore alternative scoring methods to capture a more detailed spectrum of disease impact.Comparative Analysis with Known New Zealand QoL Data: Our findings, particularly the lower QoL scores in comparison with international studies, underscore the need for a comparative analysis with existing New Zealand QoL data. Such a comparison would not only validate our findings but also situate them within the unique healthcare and demographic landscape of New Zealand, offering vital insights for healthcare providers and policymakers.

In summary, this study reveals a concerning decrease in quality of life (QoL) among patients with eosinophilic esophagitis (EoE) in New Zealand, corroborating trends noted in similar research. Elevated Dysphagia Symptom Questionnaire (DSQ) scores in our cohort suggest that current treatment strategies might not be fully addressing the symptom burden. The apparent link between DSQ scores and lowered QoL indicates a need for a review of treatment approaches. The notable occurrence of food bolus incidents points towards potential gaps in the existing management of EoE. Considering the increasing prevalence of EoE, our findings highlight the importance of refining treatment protocols to better address patient needs. The goal should be to enhance the overall QoL for patients with EoE in New Zealand, with a focus on more effective management of the disease’s symptoms and complications.

## 5. Conclusions

The QoL of patients with active EoE disease in New Zealand is reduced. The use of PPI is associated with inactive disease and non-impaired QoL. There is a high prevalence of patients with active disease and additionally a higher-than-expected prevalence of food bolus obstruction in this group.

## Figures and Tables

**Figure 1 nutrients-16-03437-f001:**
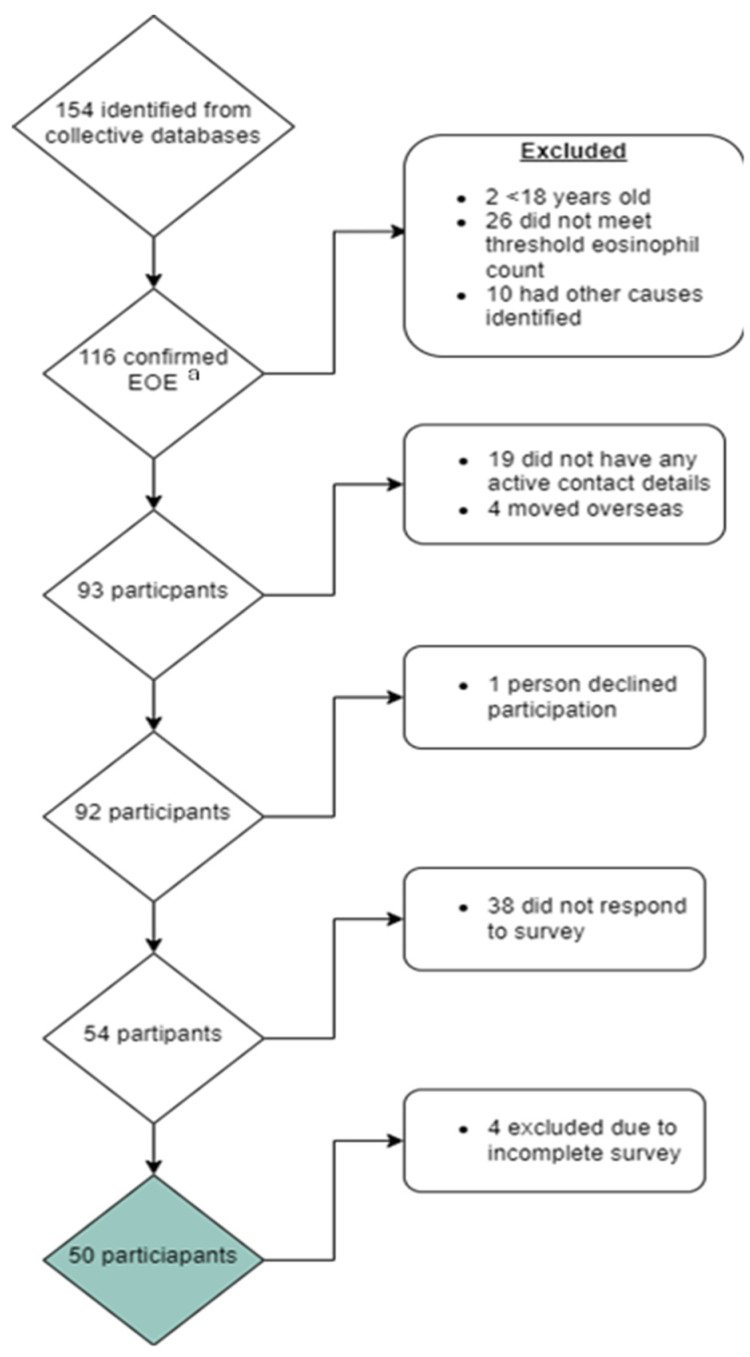
Consort diagram reporting those approached and included for participation in this study exploring quality of life in patients with eosinophilic oesophagitis (EoE). ^a^ Eosinophilic oesophagitis. Green box: Total number of final participants included in study.

**Figure 2 nutrients-16-03437-f002:**
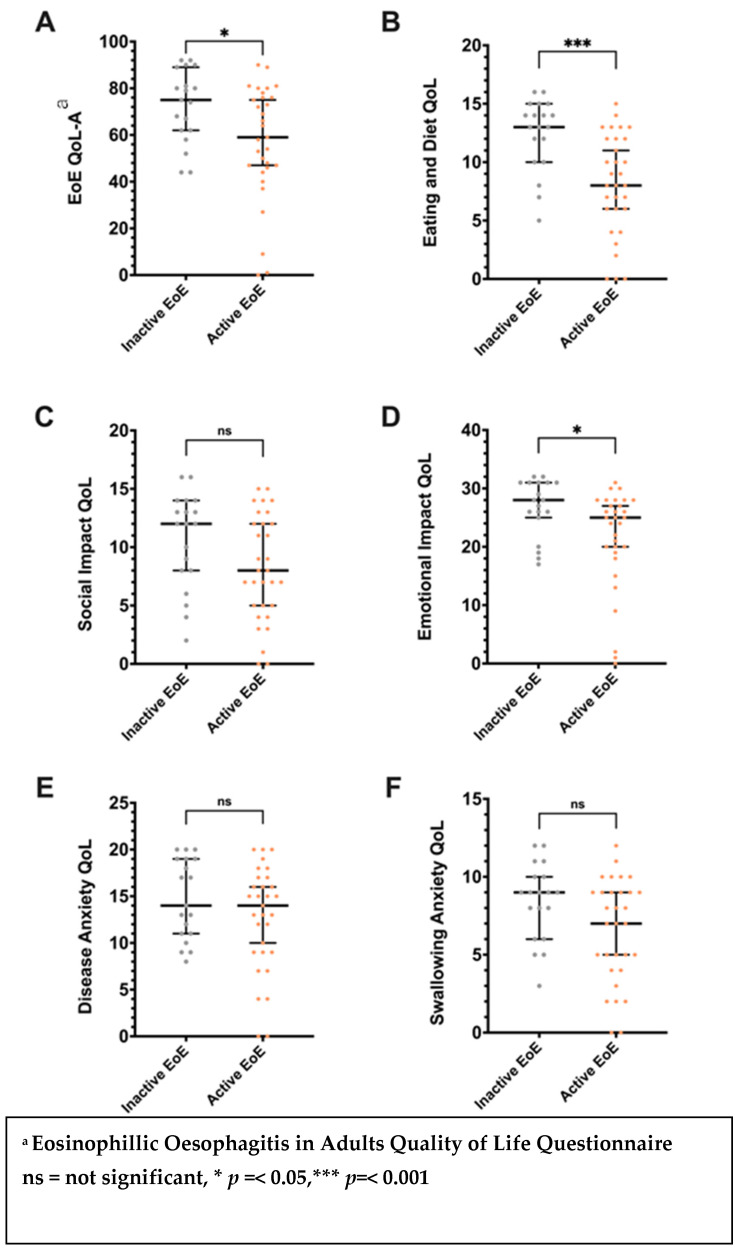
Differences in the Adult Eosinophilic Oesophagitis Quality of Life (EoE-QOL-A) questionnaire total (**A**) and subscale scores: (**B**), eating/diet; (**C**), social impact; (**D**), emotional impact; (**E**), disease anxiety; (**F**), Swallowing Anxiety; Error bars signify median and interquartile range. Associations between study variables and eosinophilic oesophagitis quality of life.

**Table 1 nutrients-16-03437-t001:** Score ranges for the total and sub-sales of the Adult Eosinophilic Oesophagitis Quality of Life (EoE-QOL-A) questionnaire.

	Score Range
Overall Score	0–96
Eating/Diet Impact	0–16
Social Impact	0–16
Emotional Impact	0–32
Disease anxiety	0–20
Swallowing anxiety	0–12

**Table 2 nutrients-16-03437-t002:** Descriptive characteristics of cohort of patients with eosinophilic oesophagitis in New Zealand. DSQ, Dysphagia Symptom Questionnaire; EoE, eosinophilic oesophagitis; EoE-QOL-A, Adult Eosinophilic Oesophagitis Quality of Life questionnaire; QoL, quality of life.

Characteristics	Total Participants (*n* = 50)
Median Age (IQR)	47 years (42–58)
Female participants (%)	11 (22)
European participants (%)	41 (82)
Previous food bolus admissions (%)	33 (66)
Steroid use (oral or topical; %)	7 (14)
Proton pump inhibitor use (%)	30 (60)
Six food elimination diet (%)	3 (6)
Median DSQ (IQR)	49 (0–60)
Active EoE (DSQ ≥ 14, %)	31 (62.0)
Median EoO-QOL-A (IQR)	68 (48–80)
Impaired QoL, EoE-QOL-A < 68 (%)	25 (50.0)
Median eating and diet QoL (IQR)	10 (7–13)
Median social QoL (IQR)	9 (5–13)
Median emotional QoL (IQR)	26 (20–28)
Median disease anxiety QoL (IQR)	14 (10–18)
Median swallowing anxiety QoL (IQR)	8 (5–9)

**Table 3 nutrients-16-03437-t003:** Differences in Dysphagia Symptom Questionnaire (DSQ) scores between groups characterised by baseline study variables.

	Median DSQ Score Group 1	Median DSQ Score Group 2	*p*-Value
Gender (Male = 1, female = 2)	49 (0–56)	56 (14–70)	0.29
Atopy/allergy/asthma (No = 1 vs. Yes = 2)	49 (0–56)	28 (0–70)	1
Oral/topical steroid use (No = 1 vs. Yes = 2)	42 (0–56)	70 (56–84)	0.02
Proton pump inhibitor use (No = 1 vs. Yes = 2)	56 (42–70)	7 (0–56)	0.01
History of food bolus obstruction (1 = No 2 = Yes)	66 (0–63)	42 (0–63)	0.86

**Table 4 nutrients-16-03437-t004:** Correlations between disease activity (Dysphagia Symptom Questionnaire, DSQ) in eosinophilic oesophagitis (EoE) and quality of life (Adult EoE Quality of Life questionnaire, EoO-QOL-A, and its subscales).

All Participants (*N* = 50)						
	DSQ Score	EoE-QOL-A	Eating/Diet	Social Impact	Emotional Impact	Disease Anxiety
EoE QoL	−0.37 **					
Eating/diet	−0.54 ***	0.87 ***				
Social impact	−0.30 *	0.90 ***	0.77 ***			
Emotional impact	−0.44 **	0.94 ***	0.82 ***	0.82 ***		
Disease anxiety	−0.20	0.84 ***	0.65 ***	0.67 ***	0.74 ***	
Swallowing anxiety	−0.19	0.80 ***	0.66 ***	0.72 ***	0.67 ***	0.63 ***

Ns = non-significant * = *p* ≤ 0.05, ** = *p* ≤ 0.01, *** = *p* ≤ 0.001.

**Table 5 nutrients-16-03437-t005:** Differences in the Adult Eosinophilic Oesophagitis Quality of Life (EoE-QOL-A) scores between groups characterised by baseline study variables.

	Median EoE-QOL-A (IQR) Group 1	Median EoE-QOL-A (IQR) Group 2	*p*-Value
Gender (Male = 1 vs. female = 2)	70 (51–81)	58 (40–75)	0.11
Comorbidities (No = 1 vs. Yes = 2)	71 (50–80)	67 (46–80)	0.86
Atopy/allergy/asthma (No = 1 vs. Yes = 2)	71 (50–80)	60 (45–84)	0.56
Oral/topical steroid use (No = 1 vs. Yes = 2)	69 (50–80)	54 (47–76)	0.34
Proton pump inhibitor use (No = 1 vs. Yes = 2)	60 (45–74)	75 (53–83)	0.02
Active disease (No = 1 vs. Yes = 1)	75 (62–89)	59 (46–76)	0.02
History of food bolus obstruction (1 = No 2 = Yes)	66 (49–79)	73 (48–81)	0.71

**Table 6 nutrients-16-03437-t006:** Univariable logistic regression analyses of the association of baseline study variables with an impaired quality of life (QoL) in eosinophilic oesophagitis. Impaired QoL determined by the presence of Adult Eosinophilic Oesophagitis QoL questionnaire score < 68. DSQ, Dysphagia Symptom Questionnaire.

	Odds Ratio	95% CI
Age	0.97	0.93–1.01
Female	1.96	0.49–7.87
Presence of any comorbidities	1.38	0.45–4.20
History of atopy	3.63	0.66–20.12
Steroids use	2.88	0.50–16.48
Proton pump inhibitor use	0.36	0.11–1.16
Food elimination use	0.48	0.04–5.65
DSQ score	1.02	1.002–1.04

## Data Availability

The raw data supporting the conclusions of this article will be made available by the authors on request.

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
