# Peer review of "Quality of Life Factors in Adults with Eosinophilic Oesophagitis in New Zealand"

_nutrients, 2024, doi:10.3390/nu16203437_

Round 1

Reviewer 1 Report

Comments and Suggestions for Authors

The research question needs to be presented more clearly.

The review of literature needs to be presented more organized manners.

Acronyms need to be explained in the figure using note.

Authors need to present the research hypotheses more clearly related to the logistic regression. Authors need to explain about variables more in the method section. 

The hypotheses development process needs to be presented in the review of literature section to make the paper more logically.

The abstract format is not suitable as compare to the journal guideline.

If the authors aimed to test the difference for variables, related hypotheses need to be presented. 

Author Response

Dear Reviewer, 

Many thanks for you time and effort spent reviewing this manuscript. Thankyou for the comments in order to improve my article.  Please see my responses below and changes made within the manuscript.

Comment 1: The research question needs to be presented more clearly.

Response 1: Updated introduction to present study question more clearly. Please see last paragraph page 2.

Comment 2: The review of literature needs to be presented more organized manners.

Response2: Please see introduction which I hope appears to be written in a more organised fashion. Page 2.

Comment 3: Acronyms need to be explained in the figure using note.

Response 3: Acronyms changed to note in Fig1 (page 7) and Fig2 (page 9)

Comment 4: Authors need to present the research hypotheses more clearly related to the logistic regression. Authors need to explain about variables more in the method section. 

Response 4: Updated introduction to present question more clearly. Please see page 2.

Comment 5: The hypotheses development process needs to be presented in the review of literature section to make the paper more logically.

Response 5: Updated introduction to present question more clearly. Please see page 2.

Comment 6: The abstract format is not suitable as compare to the journal guideline.

Response 6: Abstract modified in keeping with nutrients information to authors guideline

Comment 7: If the authors aimed to test the difference for variables, related hypotheses need to be presented. 

Response 7: Variables of active and inactive EoE addressed in hypothesis page2.

Reviewer 2 Report

Comments and Suggestions for Authors

This study assessed the determinants of quality of life (QoL) in New Zealand patients with eosinophilic esophagitis (EoE) using a questionnaire and compared with international trends. Findings highlight the importance of improving treatment options to better meet patient needs.Overall, the results were interesting, however, there are some problems that must be addressed. The manuscript needs to be carefully edited. The special comments were shown below:

1. It is recommended that the authors add relevant information such as the number of patients who participated in the interview in the abstract.

2. In the introduction, the author claims that "Previous studies have shown significant correlations be- tween EoE QoL and the degree of symptoms, and endoscopic and histological disease severity", where is the innovation of this article.

3. Similarly, in the materials section, the author lacks description of specific information about the subjects, which is important for the results of a questionnaire survey.

4. Table 2 and Figure 1 do not show that there is no difference in the impact of gender, but it appears in the description of the results in 3.1. Please explain.

5. In Table 5, what is the reference standard for Dysphagia symptom questionnaire (DSQ) score?

6. The discussion section in the article is too long. It is recommended to shorten it based on the results of the article.

Comments on the Quality of English Language

none

Author Response

Dear Reviewer, 

Many thanks for your time and effort in reviewing this manuscript. Thankyou for your comments in order to improve this article.

Comment 1: It is recommended that the authors add relevant information such as the number of patients who participated in the interview in the abstract.

Response 1: Please see results section of abstract amended, page 1. There were 93 participants contacted, 92 of which were agreeable to participate. 54 responded to survey and 50 included, 38 due to lack of response, 4 were excluded due to incomplete survey. To clarify, by interview I interpreted this to mean the survey questionnaire as no interview, please do reach out if you meant something other than the survey.

Comment 2: In the introduction, the author claims that "Previous studies have shown significant correlations be- tween EoE QoL and the degree of symptoms, and endoscopic and histological disease severity", where is the innovation of this article.

Response 2: Below are a few of the articles showing association between EoE QoL and the degree of symptoms, and endoscopic and histological disease severity

Articles showings association between EoE Qol and disease severity

a. Systematic review: health-related quality of life in children and adults with eosinophilic oesophagitis instruments for measurement and determinant factors (Reference 4)

  • This involved 34 studies; All of the papers coincide in showing that symptom severity is a major determinant of HRQoL in adult with EoE: EoE-QoL-A scores negatively correlated with dysphagia severity and with disease duration

b.  Eosinophilic oesophagitis: relationship of quality of life with clinical, endoscopic and histological activity ( Reference 15)

  • Prospective study including 99 patients. Assessed using EoE-QoL-A and an EoE activity index score followed by gastrscopy and biopsy. QoL strongly correlated with symptom severity and the presence of strictures, rings, exudates and furrows were significantly associated with lower QoL.

Papers showing association between symptoms score and histological disease severity

a. Psychometric validation of the Dysphagia Symptom Questionnaire in patients with eosinophilic esophagitis treated with budesonide oral suspension (Reference 31)

  • The was a reduction in mean DSQ score from baseline to the end of the 12-week treatment period, which correlated with histologic response. Patients with a histologic response exhibited a greater improvement in DSQ scores from baseline; mean change (±SD): −16.2 (±14.3); p < 0.0001 than those who did not have a histologic response

b. Oral Viscous Budesonide Is Effective in Children With Eosinophilic Esophagitis in a Randomized, Placebo-Controlled Trial (Dohil)

  • This was an RCT of the use of oral viscous budesonide. Primary outcomes were oesophageal eosinophilia, secondary outcomes were symptom score and histological score. This showed treatment of EoE caused significant reduction in oesophageal eosinophilia, symptom score and histological scores. “OVB is an effective treatment of pan-esophageal eosinophilia in patients with EoE, and reduction in eosinophil count (to <6 eos/hpf) correlates well with symptomatic and endoscopic improvement”

Comment 3: Similarly, in the materials section, the author lacks description of specific information about the subjects, which is important for the results of a questionnaire survey.

Response 3: Please see methods section for updated information about subjects and their recruitment. Page 3,4

Comment 4: Table 2 and Figure 1 do not show that there is no difference in the impact of gender, but it appears in the description of the results in 3.1. Please explain.

Response 4: Table 2 is descriptive statistics only.

- Figure 1 does not relate to gender however looks at the QoL scores nad the subscales between active and inactive disease but doe snot distinguish by gender.

- Table 4 shows no difference in median EoE-QoL-A between males and females, p =0.11

- Table 5 shows no difference in DSQ scores between males and females p=0.29

- Table 6 shows no difference in EoE-QoL-A using logistic regression analysis ( Confidence Interval includes 1)

Comment 5: In Table 5, what is the reference standard for Dysphagia symptom questionnaire (DSQ) score?

Apologies the description of active and inactive were mis-represented on the previous draft. I have updated our DSQ scores represented as active and inactive. This does not change stats as each category was correctly attributed to active/inactive. The mistake was taken from the fact that a daily DSQ ≥1 is considered to have symptoms but the overall score is based on their 14 score.

Our threshold was in fact 14 not 1 as previously written. Daily DSQ score reference range is 0-6. With score cumulative of 14 days (0-84).

Score ≥14 was considered active disease

There are no specified DSQ score for active/ inactive and we felt these threshold were appropriate based on  validation study in which a reduction of 13.5 points resulted in significant improvement in symptoms.

Comment 6: The discussion section in the article is too long. It is recommended to shorten it based on the results of the article.

 Response 6: Discussion section shortened. Please see page 13,14

Round 2

Reviewer 1 Report

Comments and Suggestions for Authors

Authors still present no hypotheses from rigorous review of literature. Because this paper presented the logistics reg model. It should be justified. Otherwise, the paper is not logically sound.

Author Response

Dear Reviewer, 

Many thanks for your comments in order to improve this manuscript. 

Comment1: Authors still present no hypotheses from rigorous review of literature. Because this paper presented the logistics reg model. It should be justified. Otherwise, the paper is not logically sound.

Reply1: Please see page 2 and 3, end of introduction which presents factors relating to EoE in literature other than disease and how this related to our LGA. Methods relating to dichotimisation of QoL scores addressed in methods. 

Reviewer 2 Report

Comments and Suggestions for Authors

none

Comments on the Quality of English Language

none

Author Response

Dear Reviewer, 

Many thanks for your comments in order to improve this manuscript. 

Comment1: Results must be made clearer

Reply: Please see amendments to results, page 6-15.